# Predation and spatial connectivity interact to shape ecosystem resilience to an ongoing regime shift

Agnes B. Olin [1,5] ✉, Ulf Bergström [2], Örjan Bodin [3], Göran Sundblad [2], Britas Klemens Eriksson[4], Mårten Erlandsson [2], Ronny Fredriksson[2] & Johan S. Eklöf [1]

Ecosystem regime shifts can have severe ecological and economic consequences, making it a top priority to understand how to make systems more resilient. Theory predicts that spatial connectivity and the local environment interact to shape resilience, but empirical studies are scarce. Here, we use >7000 fish samplings from the Baltic Sea coast to test this prediction in an ongoing, spatially propagating shift in dominance from predatory fish to an opportunistic mesopredator, with cascading effects throughout the food web. After controlling for the influence of other drivers (including increasing mesopredator densities), we find that predatory fish habitat connectivity increases resilience to the shift, but only when densities of fish-eating top predators (seals, cormorants) are low. Resilience also increases with temperature, likely through boosted predatory fish growth and recruitment. These findings confirm theoretical predictions that spatial connectivity and the local environment can together shape resilience to regime shifts.

Seemingly unrelated dynamic processes such as financial crises[1], social revolts[2], and power grid failures[3] can all be viewed as cascading shifts where an alternative state rapidly spreads through a system. The dynamics of these shifts generally depend on the stressor (i.e., the external driver of change), each individual unit's ability to resist and recover from shifts, and, finally, the connectivity between units[4]. While strong connectivity can in some cases facilitate the spread of the alternative state (e.g., a viral disease spreading rapidly through a well-connected social network), it can, importantly, also enhance resilience to the shift (e.g., the same network acting to spread information and norms regarding preventative measures[5]).

Similar dynamics also play out in ecosystems that display alternative regimes. Here, an abrupt disturbance or a gradual change in conditions can move the system across a tipping point and into a new, persistent regime with different species compositions and dynamics[6]. In large, heterogeneous ecosystems, regime shifts may occur through asynchronous local shifts, which appear gradual at the whole-system-level—so-called spatial or gradual regime shifts[7,8]. Processes that link close-by areas (e.g., the movement of species, the flow of water) will affect the dynamics of the shift, and may generate process-specific patterns such as propagating fronts[7]. Strong spatial connectivity could thus aid the spread of the alternative regime. However, theory suggests that spatial connectivity could also provide resilience to the shift, for example by benefitting the spread and persistence of organisms that uphold the original regime[9,10]. This resilience can be in the shape of increased resistance (e.g., a population boosted by immigration, thus resisting a shift) and/or recovery (e.g., recolonisation of a species lost in a local shift), both of which are important components of ecosystem resilience[11]. While strengthening habitat connectivity is already an increasingly employed strategy within habitat and species conservation[12,13], much less is known about the role that spatial connectivity could play in preventing and reversing regime shifts, and how

[1]Department of Ecology, Environment and Plant Sciences, Stockholm University, Stockholm, Sweden. [2]Department of Aquatic Resources, Swedish University of Agricultural Sciences, Uppsala, Sweden. [3]Stockholm Resilience Centre, Stockholm University, Stockholm, Sweden. [4]Groningen Institute for Evolutionary Life Sciences, University of Groningen, Groningen, the Netherlands. [5]Present address: Department of Aquatic Resources, Swedish University of Agricultural Sciences, Uppsala, Sweden. ✉e-mail: agnes.olin@slu.se

this effect may interact with local environmental drivers. Such knowledge is crucial in light of the often large ecological and economic consequences of regime shifts (see ref. 14). So far, however, existing work is mostly limited to theoretical and conceptual studies (e.g., refs. 8,9,15. but see e.g., ref. 16), and there is a clear need for more empirical studies of how these dynamics play out in real ecosystems.

Here, we explore the roles of connectivity and the local environment in driving resilience to a spatially propagating regime shift in a large, heterogeneous ecosystem—the Swedish coast of the Baltic Sea, a brackish arm of the Atlantic Ocean. Along this coastline, the resident predatory fish species European perch (*Perca fluviatilis*) and northern pike (*Esox lucius*) are increasingly being replaced by one of their prey species, the three-spined stickleback (*Gasterosteus aculeatus*)—a small,

opportunistic, mesopredatory fish[17–19]. Even though these species share similar preferences for spawning in shallow bays distributed throughout the archipelago, the bays tend to be completely dominated either by stickleback or by their predators[17,20,21] (Fig. 1a). Multiple independent field and experimental studies suggest that this dichotomy is driven by predator-prey reversal (Fig. 1a): at high densities, adult predators suppress stickleback abundances through predation, while at low predator densities, the stickleback are released from top-down control, and instead suppress predator recruitment by preying on the predators' eggs and larvae[17,22,23]. A shift to stickleback dominance is not only detrimental to pike and perch populations, but also has cascading effects on lower trophic levels (Fig. 1a), resulting in higher densities of filamentous algae, which reduces water and habitat

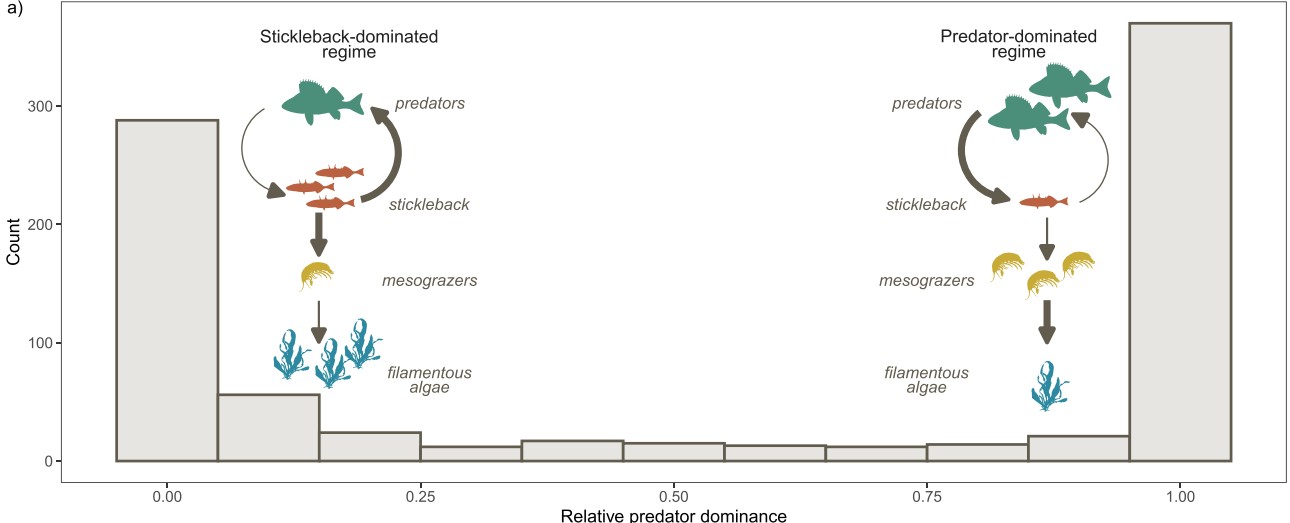

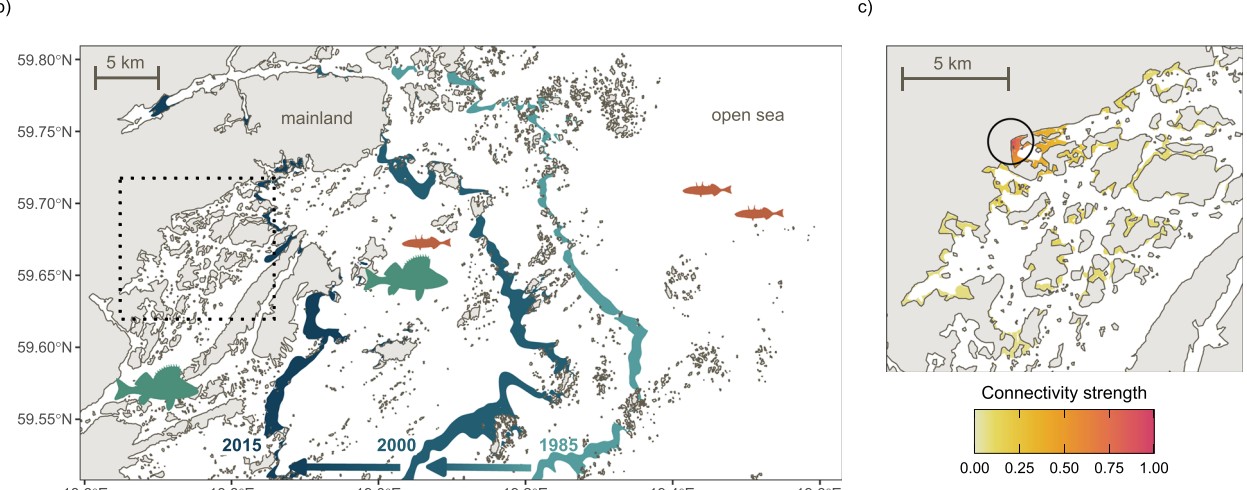

**Fig. 1 | A spatially propagating shift in dominance from predatory fish to stickleback. a** Histogram of relative predator dominance (density of predators/ density of predators and stickleback) based on juvenile data collected 1979–2020 along the Swedish Baltic Sea coast (see ref. 17), demonstrating the strong bimodality of the system, together with illustrations of the feedback mechanisms that drive the system towards a predator- or stickleback-dominated state, and the resulting top-down cascade. Arrow thickness indicates strength of interaction. The mesograzer and algae symbols are courtesy of Tracey Saxby, the Integration and Application Network (ian.umces.edu/symbols/). **b** An illustration of the spatially propagating regime shift, where the stickleback-dominated regime has gradually expanded from the outermost islands in the archipelago towards the mainland. Coloured zones represent a spatial threshold of predicted equal probability of

stickleback and predator dominance for three points in time (1985, 2000, 2015) based on a generalised linear model fitted to the data in 1a (see ref. 17). Arrows show how the stickleback wave has moved towards the mainland over time. The dashed box shows the extent of 1c. **c** Illustration of the spatial scale of predator habitat connectivity, where coloured areas indicate predator spawning habitat defined based on environmental preferences (see Methods). The colour scale indicates strength of dispersal connectivity from a focal point (empty circle), with the probability of dispersal declining rapidly as a function of distance, based on empirical data (Supplementary Fig. 1). Source data are provided as a Source Data file for (**a**). Coastline in (**b**) and (**c**) from the European Environment Agency (eea.europa.eu/ds_resolveuid/78DY1XZFJ2).

quality[24–26]. Over the past few decades, the stickleback-dominated regime has gradually expanded from areas close to the open sea and further into the archipelago, forming a spatially propagating regime shift nicknamed the stickleback wave[17] (Fig. 1b). The underlying trigger of the regime shift is unknown, but is most likely a combination of i) increasing stickleback abundances, likely primarily driven by reduced predation in their offshore habitat[27] and ii) a loss of resilience of the predator populations due to habitat degradation, fishing, and predation from great cormorant *Phalacrocorax carbo* and grey seal *Halichoerus grypus*[28].

The clear bimodality of stickleback- and predator-dominance (Fig. 1a), combined with a strong predator-prey reversal feedback mechanism (which can underpin alternative stable states[29,30]) and observed persistent flips from predator- to stickleback-dominance in single locations[17], together suggest that a shift to stickleback dominance could represent a critical transition to a new, stable state. In any case, the stickleback wave clearly falls within the broader definition of regime shifts as "dramatic, abrupt changes in the community structure that are persistent in time, encompass multiple variables, and include key structural species"[31].

With access to a large dataset across a highly heterogeneous habitat (>7000 samplings of the juvenile fish community in an island-rich archipelago covering ca. 680 km of the Swedish Baltic Sea coast over two decades), the stickleback wave provides a good model system for empirically assessing the role of connectivity and the local environment in shaping resilience to regime shifts. Motivated by species-specific studies[32], as well as wider theory and empirical studies suggesting that occupancy and biomass of a habitat patch increases as a function of connectivity[16,33–35], we expect strong connectivity with nearby predator spawning habitat to result in higher local densities of adult predators. As previous studies show that high predator densities confer resilience to a shift to stickleback dominance[17,19,24], we thus expect connectivity to increase resilience to the stickleback wave. While pike and perch are coastal residents throughout the year with highly localised movement (typically dispersing <10 km[36]; Fig. 1c), we do not expect stickleback to depend on local connectivity in the same way. Instead, the stickleback population is connected over vast distances (the Swedish Baltic Sea coast constitutes a single genetic cluster[37]), with mature stickleback migrating long distances to the coast from the open sea at the time of spawning, and juveniles and surviving adults migrating offshore at the end of summer[17,20]. The amount of immigrating stickleback can thus be considered as a stressor gradient through the archipelago, dependent primarily on the distance from the open sea and the density of stickleback offshore.

In line with theory[8,15], we also expect the dynamics of the ongoing regime shift to depend on local drivers of resilience (i.e., environmental drivers that modify the predators' ability to resist and/or recover from a local flip to stickleback dominance). Here, we consider two key variables known to locally suppress adult predatory fish: i) predation from seals and cormorants[38–40] and ii) fishing[41,42]. We also expect the effects of fishing and predation to interact with the effect of connectivity. Our measure of connectivity by itself (see "Methods") only captures potential connectivity (i.e., distance to, and area of, nearby predator spawning habitat patches), which is unlikely to have any effect on local predatory fish densities (and thus resilience) if there are no fish to move between habitat patches. Including this interaction between connectivity and predation/fishing allows us to at least partially account for the availability of predatory fish that can move between patches, assuming that an effect of fishing and/or predation on predatory fish densities is present in the dataset. We thus expect a strong positive effect of connectivity at low predation and/or fishing pressure (as there are more predatory fish to redistribute), which then weakens with increasing extraction of predatory fish. In addition, we also consider the effect of water temperature, hypothesising that warmer spawning seasons increase the resilience of the predatory fish

by increasing the growth rate of their larvae (see ref. 43), effectively advancing and shortening the period during which they are sensitive to stickleback predation (see refs. 21–23), while at the same time increasing the predators' consumption rate of stickleback (see ref. 44). Here, we expect a stronger positive effect of temperature close to the open sea, where stickleback probably arrive both earlier and in greater numbers, making fast and early predator larvae growth more important.

## Results

### Effects of incoming stickleback

First, we established a baseline model including variables representing the amount of incoming stickleback to a given location, i.e., the stressor of the system. This was done using generalised linear mixed models with spatio-temporal random effects ($N = 3491$), with predatory fish dominance as the response variable (density of pike and perch juveniles divided by density of pike, perch and stickleback juveniles; 0 = complete stickleback dominance, 1 = complete predator dominance; see ref. 17). The data, collected in 2001–2020 at the end of summer along the Swedish Baltic Sea coastline at latitudes 55–60.5°N, only include juveniles and thus reflect the outcome of interactions during the spawning season, which generally mirrors the relative dominance of spawning adults in spring[17].

As expected, predatory fish dominance increased with distance to the open sea (i.e., with longer stickleback migration distances), and decreased both with the estimated density of mature stickleback offshore (a proxy for the amount of incoming spawners; measured using acoustic surveys in autumn, see ref. 45) and with wave exposure (a measure of the openness of an area and thus the accessibility for stickleback) (Supplementary Fig. 2; Supplementary Table 1). The strong effect of distance to the open sea mirrors previous findings[17], but here we could also link the spatially propagating regime shift to offshore stickleback densities. The dynamics of the absolute densities of juvenile predators ($N = 7415$) and stickleback ($N = 7167$) largely reflected the dynamics of their relative dominance (Supplementary Tables 2, 3).

### Effects of connectivity, top predators, fishing, and temperature

To explore the role of connectivity and local environmental drivers (predation pressure, fishing pressure, temperature) in providing resilience to this spatially propagating regime shift, we used generalised linear mixed models with a random effect of year. Connectivity was calculated based on predator spawning habitat maps produced using cut-offs for depth and wave exposure, assuming distance-dependent dispersal (Fig. 1c; see Methods for details). The models also included interactions between connectivity and fishing/predation and between temperature and distance from the open sea, as described in the Introduction, as well as variables related to the amount of incoming stickleback (distance to open sea, offshore stickleback densities, wave exposure).

Resilience is inferred from a positive effect of a given driver on the probability of predator dominance, so that for the same level of disturbance (i.e., constant values of the variables representing the amount of incoming stickleback), the system is more likely to be predator-dominated at higher values of the driver. This could result both from a given driver increasing resistance to the stickleback wave (so that a location remains predator-dominated at the time of sampling), and from a given driver facilitating recovery (so that a location has again become predator-dominated at the time of sampling). While we are not able to distinguish between the two mechanisms, they are both important components of resilience[11], and our study therefore provides useful information on the functioning of the ecosystem, as well as on effective management measures. Since the density of adult predators (which we do not have data for) regulates an area's ability to suppress stickleback, a positive effect on the absolute densities of

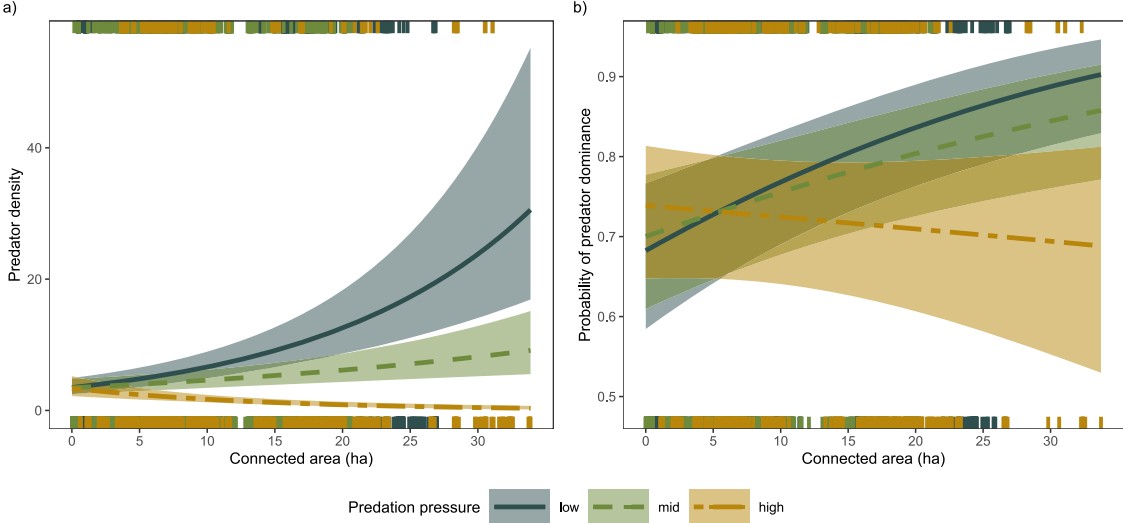

**Fig. 2 | Effects of connectivity and predation pressure on the densities and dominance of predatory fish. a** Predatory fish densities and **b** probability of predator dominance as a function of predatory fish spawning habitat connectivity for different levels of seal and cormorant predation pressure. low = 10th percentile, mid = median, high = 90th percentile. Lines show predictions from a generalised linear mixed model including a random effect of year and other variables representing incoming stickleback (offshore stickleback, distance from open sea, wave exposure) and local environment (temperature, fishing), with 95% confidence intervals. Notches show distribution of raw data. Predator densities refer to the number of individuals in a sampling area of roughly 80 m². Source data are provided as a Source Data file.

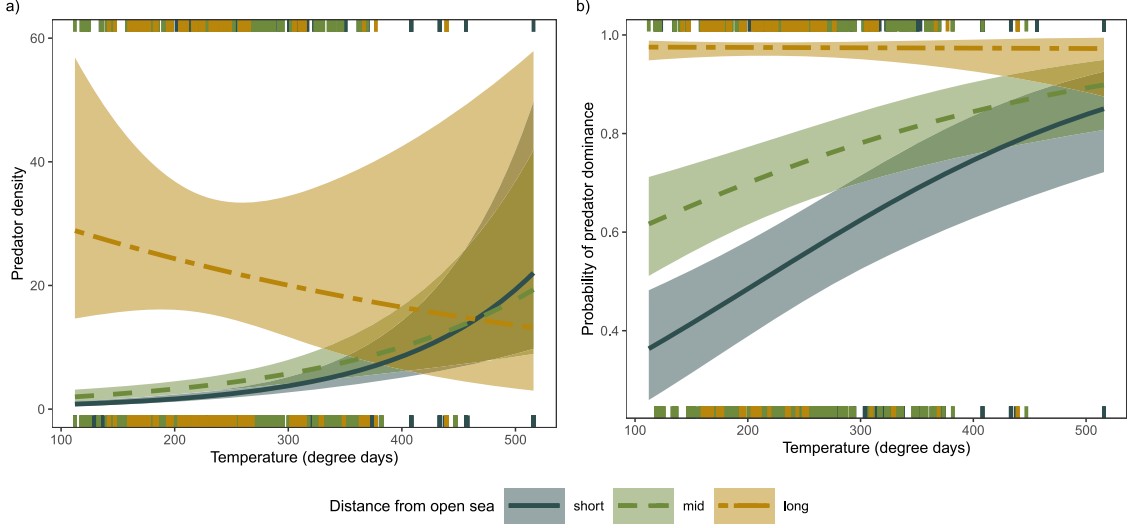

**Fig. 3 | Effect of temperature on the densities and dominance of predatory fish. a** Predatory fish densities and **b** probability of predator dominance as a function of temperature (summed degree-days over 10 °C) for different distances from the open sea. short = 10th percentile, mid = median, long = 90th percentile. Lines show predictions from a generalised linear mixed model including a random effect of year and other variables representing incoming stickleback (offshore stickleback, wave exposure), connectivity and local environment (predation, fishing), with 95% confidence intervals. Notches show the distribution of raw data. Predator densities refer to the number of individuals in a sampling area of roughly 80 m². Source data are provided as a Source Data file.

predator juveniles can also be interpreted as increasing resilience, as good recruitment is a prerequisite, if not a guarantee, for a strong adult population.

We found support for an effect of connectivity on both predatory fish dominance and absolute predatory fish densities (Fig. 2; Supplementary Tables 4–6), as well as an interaction between connectivity and predation pressure from seals and cormorants (calculated based on counts of seals and cormorants combined with estimates of foraging range and consumption rates). Connectivity had a positive effect at low and medium predation pressure, but the effect disappeared (Fig. 2b), or even turned negative (Fig. 2a) at high predation pressure.

For stickleback densities, there was no clear effect of connectivity, but a negative effect of predation pressure (Supplementary Table 6; Supplementary Fig. 3). Moreover, fishing pressure (commercial catch data combined with recreational catches estimated using questionnaires) had a negative effect on predatory fish dominance, but there was no support for an effect on absolute predatory fish or stickleback densities, and no support for an interaction with connectivity (Supplementary Tables 4–6; Supplementary Figs. 4, 5).

Finally, we also found a positive effect of spawning season temperature on predator dominance and predatory fish densities (Fig. 3; see Supplementary Tables 4–6), where local temperature

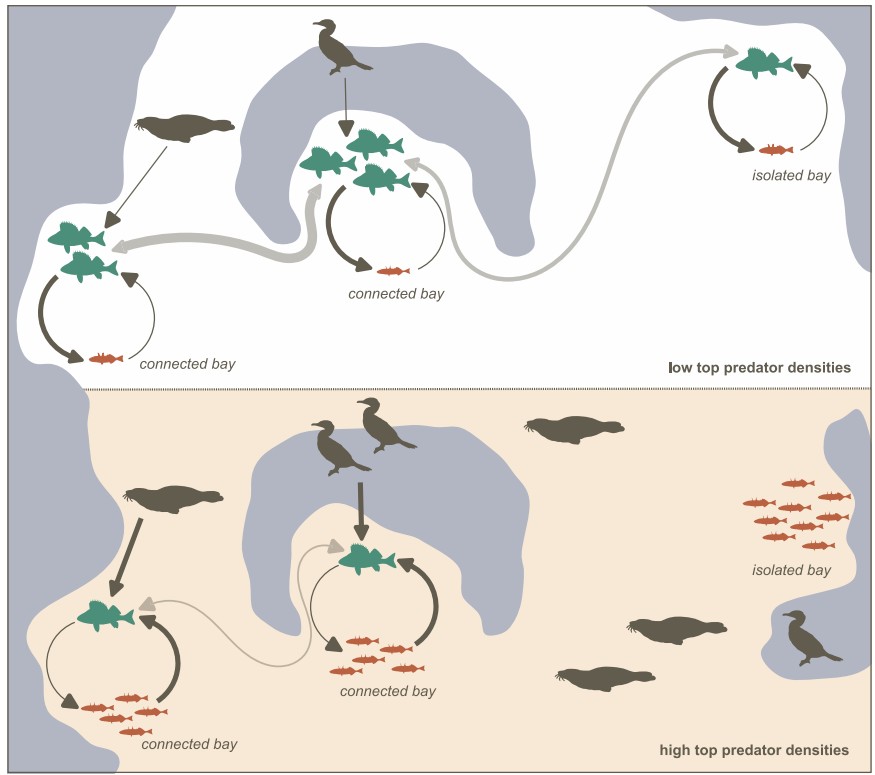

**Fig. 4 | Illustration of the identified interactive effects of connectivity and predation pressure.** Dark grey arrows indicate predation effects, with line thickness indicating the strength of the effect. Light grey arrows indicate dispersal, with line thickness indicating the relative abundance of dispersing predatory fish. The seal and cormorant symbols are courtesy of Diana Kleine, Kim Kraeer and Lucy Van Essen-Fishman, the Integration and Application Network (ian.umces.edu/symbols/).

development was calculated based on satellite data as the degree day sum of daily mean temperatures >10 °C from the beginning of the year until the start of juvenile data collection (temperatures >10 °C provide beneficial predator spawning and growth conditions[43]). For absolute predatory fish densities, there was also support for an interaction between temperature and distance to the open sea, where warm water temperatures resulted in higher predator densities close to the open sea, but had no clear effect in the inner archipelago (Fig. 3a). For stickleback densities, the support for a temperature effect was more inconsistent (Supplementary Table 6), with a possible effect largely opposite to the relationships observed for predatory fish densities (Supplementary Fig. 6).

**Relative importance of different drivers**

Examining the amount of variance explained by the different drivers can provide important clues to the dynamics of the shift (see Supplementary Table 7). We first compared the amount of variance explained by the fixed effects in the baseline model including only variables related to the amount of incoming stickleback ($R^2_{baseline}$), i.e., representing only the stressor in the system, versus the model also including connectivity, all local environmental variables and interactions ($R^2_{full}$), i.e., variables which may modify resilience to this stressor. The variance explained by these two models were similar for predator dominance ($R^2_{baseline} = 0.34$, $R^2_{full} = 0.38$) and stickleback densities ($R^2_{baseline} = 0.36$, $R^2_{full} = 0.38$), but differed clearly for predatory fish densities ($R^2_{baseline} = 0.23$, $R^2_{full} = 0.38$). A model of predatory fish densities including only connectivity, local environmental variables and interactions, i.e., excluding the effect of incoming stickleback, explained roughly the same amount ($R^2 = 0.36$) as the full model ($R^2_{full} = 0.38$). Together, this can be interpreted as local, coastal dynamics being important for driving the absolute densities of predatory fish, while stickleback densities and the relative dominance are

to a larger extent driven by variables used to represent the influx of stickleback from the open sea. For absolute densities of predatory fish, dropping the effect of predation from the full model ($R^2_{full} = 0.38$) resulted in the largest decrease in amount of variance explained ($R^2 = 0.25$), followed by connectivity ($R^2 = 0.34$), and then temperature and fishing ($R^2 = 0.37$ for both). Of these variables, predation thus explains the largest amount of variation in predatory fish densities.

## Discussion

Using an extensive spatio-temporal dataset, we found that predation, connectivity, and local temperature all interacted to shape resilience to an ongoing, spatially propagating shift in dominance from large predatory fish to the opportunistic stickleback along the Baltic Sea coastline. Predator habitat connectivity increases resilience to the shift, but only when densities of fish-eating top predators (grey seals and great cormorants) are low. In addition, the ability of the predatory fish to withstand the shift increases with temperature, likely through shortening and advancing the window during which the predator offspring are sensitive to stickleback predation. As such, our study provides empirical support for both connectivity and the local environment shaping resilience to regime shifts.

The positive effect of habitat connectivity on predatory fish densities and dominance in areas with low seal and cormorant densities suggests that when predatory fish are abundant (due to low predation pressure; see refs. 38–40), the movement of adult predatory fish between spawning areas allows for redistribution and recolonisation, increasing local resilience (Fig. 4). In contrast, this positive effect of connectivity was absent in areas with high predation pressure from top predators, most likely because there were too few adult predatory fish left to redistribute. For absolute predatory fish densities, there was even a weak negative effect of high connectivity when top predator densities were high. This could possibly

be interpreted as a form of smearing effect where the few predatory fish that remain spread out, creating low density populations that are more vulnerable to the incoming stickleback. This, in line with theory[9,10] and empirical work[33], illustrates well how connectivity can have contrasting effects that depend on local conditions. It should be noted that due to the large spatial scale of the study, our connectivity metric had to be relatively coarse, not accounting for local variables affecting habitat quality (e.g., vegetation, prey availability) and dispersal paths (e.g., width of passages). That we still identified an effect, even if the additional variation explained was modest, suggests that the actual effect is larger. A better understanding of what may constitute barriers to dispersal (see e.g., ref. [46]), and whether juveniles disperse similar distances as adults (see Methods), would be helpful for refining estimates of connectivity in this study system. In addition, we need a better understanding of the presence of directional movement, such as movement towards high-quality areas or away from high-density areas, which would have a large impact on which areas are losing (net emigration) vs gaining resilience (net immigration).

As seals and cormorants primarily consume larger fish[38–40], the negative influence of top predator densities on the predatory fish juveniles likely reflects a reduction in the adult population (relative predator dominance in juveniles at the end of summer generally mirror relative predator dominance in adults in spring[17]). Cormorants and seals also feed on stickleback[40,47], which may explain the negative relationship between top predators and stickleback densities found in this study. Still, the negative effect on the probability of predator dominance suggests that the net effect of top predators is to push the system towards stickleback dominance, which is in line with perch making up a larger proportion of grey seal and great cormorant diet than stickleback[47]. The large difference in the variation explained by a model of juvenile predator densities including an effect of predation ($R^2 = 0.38$) and by one excluding such an effect ($R^2 = 0.25$) suggests that predation from seals and cormorants could be an important driver of pike and perch populations (see also refs. [38–40]). Together with previous work showing that in areas dominated by stickleback, densities of grazers (amphipods, gastropods) are suppressed through stickleback predation, resulting in greater densities of filamentous algae due to low grazing pressure[25,26], our results hint at the possible presence of a five-level trophic cascade.

We found no effect of fishing on absolute predator densities or stickleback densities. A weaker effect of fishing than of seals and cormorants agrees well with previous studies suggesting that the amount of coastal predatory fish extracted by fishers has been substantially lower than the amount extracted by marine mammals and birds in recent decades[41,47]. Still, there was some support for a negative effect of fishing on the probability of predator dominance, suggesting that fishing on predatory fish could contribute to pushing an area over to stickleback dominance. However, it should be noted that our measure of fishing pressure was coarse, and likely also reflects general exploitation, which has been linked to both reduced predator habitat quality[48] and high densities of stickleback[49].

Warm water temperatures during the spawning season had a positive effect on predator dominance and densities of juvenile predators, likely through advancing and shortening the window during which the predator larvae are sensitive to stickleback predation, and possibly also by boosting predatory fish consumption rates (see ref. [44]). The effect was, as expected, stronger close to the open sea, suggesting that the benefit of fast and early growth is larger where the migrating stickleback presumably arrives earlier, and in larger numbers (see refs. [22,23]). The Baltic Sea is showing particularly fast rates of warming in response to climate change[50], but while several studies have identified increasing temperatures as a trigger for regime shifts[51,52], warming may in this case instead decrease the risk of a shift. As such, our results provide some nuance to the framing of

temperature as solely an external driver of regime shifts (see e.g., ref. [31]). However, temperature did not explain a large proportion of the variation, and our results do not exclude the possibility that warming may still benefit stickleback in offshore areas (see ref. [53]). Further work is thus needed to determine the net effect of increasing Baltic Sea temperatures on the dominance of stickleback vs predators.

Our findings also help shed light on the overall dynamics of the ongoing regime shift. Drivers related to the amount of incoming stickleback explained most of the variation in the probability of predator dominance, suggesting that the stressor (increasing stickleback densities) is the key determinant of whether the juvenile community is dominated by stickleback or predators in a given year. However, as distance from the open sea and wave exposure co-vary with variables that determine predatory fish habitat suitability (e.g., correlation between wave exposure and vegetation[24]), partitioning variation is tricky with this type of correlative data. Still, together with the demonstrated ability of stickleback to suppress predatory fish recruitment[17,22,23], our results suggest that the stickleback increase actively contributed to the ongoing regime shift (as opposed to the shift being a consequence of predator declines only). This in turn emphasises the importance of ongoing work to identify the drivers of the stickleback increase, which currently points to a key role played by population declines of stickleback predators and competitors offshore[27]. At the same time, the role of top predators, the local environment and habitat connectivity identified in this study, as well as previous work suggesting that declines in predatory fish preceded stickleback dominance in at least some locations[17,23] and that strong predatory fish populations can prevent stickleback occupation[17], support the view that the shift is driven by a combination of a loss of resilience and an increasing stickleback population. This is fully in line with most regime shifts, which usually start by a drop in resilience, allowing a stressor to then push the system over the edge[31]. The few locations in our dataset that do have repeated sampling (Supplementary Fig. 7) suggest that a flip from predator dominance to stickleback dominance is an extended process that may involve a period of flickering, as is commonly observed in regime shifts[4]. Likely, a mix of processes may be involved in this flickering period, altering the dominance dynamics between years. For example, the amount of incoming stickleback will vary from year to year as a result of e.g., overwinter survival offshore, while predator juvenile recruitment and growth rate (and thus their vulnerability to stickleback predation) may vary as a result of e.g., temperature (as found in this study). That a location can, at least temporarily, bounce back to predator dominance can be interpreted at least as potential for recovery (even if it does not represent recovery from a stable flip). At the same time, years of intermediate dominance in the course of an ongoing flip in our time series may hint at resistance. No time series show a more stable flip back to predator dominance, however, suggesting that we are not seeing any long-term recovery at this moment in time. Further studies teasing apart the processes driving both resistance and recovery would provide a better understanding of the system and could help steer future management actions.

Predatory fish dominance is typically considered the societally desired regime in our system due to their ability to counteract the effects of eutrophication, maintaining clear waters and healthy macroalgae via strong top-down effects[25,26], and their popularity as recreational fishing targets[54]. To boost the predator-dominated state, our results suggest that management efforts may be usefully aimed at reducing the amount of incoming stickleback by i) limiting stickleback population growth (e.g., by reducing fishing pressure on stickleback predators in the open sea[27]), ii) preventing access to predator spawning areas (e.g., by minimising the dredging of boating canals[21]) and/or iii) initiating an offshore fishery for stickleback in the Baltic Sea (although we strongly urge for a cautionary approach; see ref. [55]). In addition, both empirical and theoretical work suggest that boosting

the resilience of the desired state is equally, if not more, important[6], especially in a preventative strategy that considers possible hysteresis effects. Based on our findings, this could include i) managing seal and cormorant populations (e.g., reducing fishing pressure on Atlantic herring *Clupea harengus* to increase densities of alternative prey; preventing access to important predatory fish habitat by using scares or nets), ii) aiming local management efforts at areas of high connectivity, as this may multiply any positive impacts, and ensuring that dispersal pathways between spawning areas are maintained, e.g., by limiting physical disturbance of important dispersal corridors, iii) ensuring predatory fish have access to habitats with beneficial temperature development in spring (e.g., enclosed bays), and iv) reducing local fishing pressure on coastal predatory fish. The role of different drivers will vary along this heterogeneous coastline, meaning that measures need to be adapted to local conditions. Other studies also point to the importance of restoring and increasing access to freshwater spawning habitat[22,23,56], limiting physical disturbance of sensitive areas[48] and reducing eutrophication[57]. It would also be valuable to consider the role of these measures in modifying resilience, especially since considerable unexplained variation remains. As eutrophication likely benefits spawning sticklebacks[27,58] while reducing predator spawning habitat quality[57], we had also planned to include eutrophication status in the analysis, but unfortunately no suitable data were available (see further in "Methods"). Any measures carried out should be accompanied by rigorous monitoring, which would allow for local, experimental testing of the effects that we identified using large-scale correlational data.

The central aim of this study was to empirically assess the role that connectivity plays in shaping resilience to regime shifts and how this effect acts alongside local environmental conditions. To sum up, our findings support theoretical predictions[8–10,15] that, while it may not be the main driver, connectivity can boost resilience to regime shifts, and that this effect interacts with local drivers (in our case, local seal and cormorant densities). Over the last few decades, there has been considerable focus within conservation on managing for high connectivity, e.g., through reserve networks and wildlife corridors[12,13]. Our results support this focus on maintaining and rebuilding connectivity, and suggest that this can in some cases also increase resilience to regime shifts. Further, our work also highlights that local efforts, in our case managing populations of seals and cormorants, can generate positive effects that spill over to adjacent areas and thus provide benefits on a larger scale. Finally, our results clearly point to the importance of a spatial perspective for increasing the understanding of drivers of, and resilience to, regime shifts in large, heterogeneous ecosystems.

## Methods
### Juvenile fish densities
Juvenile fish density data were assembled from a range of research projects and monitoring programmes, carried out mainly by the Swedish University of Agricultural Sciences, the Swedish Board of Fisheries, Stockholm University, the county boards along the Swedish Baltic Sea coast and the consultancies/organisations Naturvatten i Roslagen AB, JP Aquakonsult, Upplandsstiftelsen, Sveriges Vattenekologer AB, Hushållningssällskapet Rådgivning Nord AB and Hydrophyta. No data were collected specifically for the present study. The majority of the data are available in the Swedish national database for coastal fish monitoring (slu.se/kul). All sampling was carried out by certified personnel with necessary permits, in compliance with the EU Directive 2010/63/EU and national legislation. The main scientific surveys included in the dataset were covered by Permit 2007-0883 issued to the Swedish Board of Fisheries by the Swedish Animal Welfare Agency and Permit C 139/13 issued to the Swedish University of Agricultural Sciences by the Ethical Committee on Animal Experiments.

All data were collected in shallow coastal areas using low-impact pressure waves that stun or kill most fish with a swim bladder within a given radius[59]. The fish, either only floating or both floating and sunken fish, are then collected and counted. As juvenile and adult sticklebacks can be difficult to tell apart in the field, these were added together (absolute majority will be juveniles at this time of the year). To be able to compare across samples, we calculated the proportions of fish floating when both floating and sunken fish were collected ($N_{perch}$ = 2735, $N_{pike}$ = 2291, $N_{stickleback}$ = 3844) and used these as correction factors in instances when only floating fish were collected. Further, we also corrected for variation in the strength of the detonations using previously calculated factors correcting to the current standard (10 g Pentex explosive), so that each value corresponds to the number of fish within roughly an 80 m² area[60].

The data covered the period from the end of July (towards end of stickleback spawning season[20]) until the end of September (peak offshore stickleback migration[20]). We subset the data to the years 2001–2020, as data were scarce prior to this, and only included data below a latitude of 60.5°N, delineating an area, the Baltic Proper, that is often considered a distinct unit from both a management and an ecosystem perspective. Finally, we removed data further than 40 km from the open sea ($N$ = 88), since areas further in were poorly sampled (out of the 42 samples with fish in them, >95% were predator-dominated, as would be expected this far into the archipelago; see also ref. 17). The data only include juveniles, and thus reflect the outcome of interactions during the spawning season. However, the relative dominance of juveniles of predatory fish and stickleback at the end of summer generally reflects the relative dominance of spawning adults in spring[17]. Existing surveys of adult fish are much more limited in both space and time, and do not include stickleback, which is why juvenile data were used.

### Incoming stickleback
The influx of stickleback to the coastal spawning areas in spring was expected to decrease with distance to the open sea, as well as to increase with the amount of mature stickleback in the open sea and the openness of the area (here represented by wave exposure). The cost distance function in ArcGIS Pro 2.4[61] and a 5 × 5 m landmask were used to calculate distance to the open sea (with the border between the archipelago and the open sea considered as a line tracing the outer islands in the archipelago; created by first expanding a land raster by 5 km and then removing the outer 5 km) (Supplementary Fig. 8). The wave exposure layer was produced using the Simplified Wave Model[62] and had a resolution of 10 × 10 m (Supplementary Fig. 9). Values for each juvenile sampling point were then extracted from these layers. As a proxy for the amount of incoming spawners (Supplementary Fig. 10), we used the distance-weighted mean biomass of potential spawners in the open sea in October the year before (individuals ≥5.5 cm, see ref. 20), based on data from the Baltic International Acoustic Survey, which combines acoustic data with trawls to estimate densities at the scale of ICES statistical rectangles (see ref. 45). The distance-weighting was centred on the archipelago-open sea border at the closest point from the focal juvenile sampling point, and used a normal distribution where ~95% of the weight was allocated to distances ≤150 km (home range of stickleback >100 km[36]). If there were <3 ICES rectangles with data within a radius of 150 km, we excluded the datapoint.

### Connectivity
As described in the Introduction, we considered connectivity as the amount of accessible neighbouring predator spawning habitat. It is in the spawning habitat that the interactions between predators and stickleback occur[17,21], and recruitment habitat area is also limiting for adult predator populations[32], which is why we focused on this habitat type. Both pike and perch have a particular preference for spawning in shallow and sheltered areas[59,63] and we, therefore, delineated spawning

habitat using cut-offs for depth (≤3 m) and wave exposure (two alternatives: ≤3.5 and ≤3.2, on a $\log_{10}$-scale; see Supplementary Fig. 11). These cut-offs were based on perch surveys in the central Baltic Sea[32,59] but are also largely sensible for pike[63]. These variables together explain ca. 50% of variation in perch egg strand occurrence[59]. This type of habitat is often found in bays (why the main text refers to spawning bays), but suitable spawning areas can also occur along more open coastlines if conditions are right. When delineating the spawning habitat, we worked with a landmask at a 5 × 5 m resolution, a depth layer of 250 × 250 m resolution (smoothed into 5 × 5 m via bilinear resampling) and a wave exposure layer[62] of 10 × 10 m resolution (also smoothed into 5 × 5 m). As very small habitat patches are unlikely to constitute productive spawning habitat, we removed patches smaller than 1 ha[63]. The habitat calculations were done in QGIS 3.18[64].

The interannual movement between patches (individuals born in bay A later spawning in bay B, or an individual spawning in bay A spawning in bay B in a later year) was assumed to be distance-dependent. We estimated the cumulative probability of dispersal for different distances by fitting an asymptotic Michaelis-Menten model to observations of adult perch tagged and re-caught during the spawning season from a tagging study in south-western Finland[65] (Supplementary Fig. 1). The extent to which this is representative of juvenile dispersal is unclear, but in comparison with adult dispersal (see Supplementary Fig. 1), estimated larval dispersal distances are generally short (0.1–2 km[36] over a duration of ca. 2 months before metamorphosis and a return to the littoral zone[66]) and studies of Baltic perch population genetics suggest that all dispersal is highly localised[67]. Pike disperse even shorter distances and exhibit strong homing behaviour, but dispersal distances are still of a similar magnitude[36]. While remaining at the coast throughout the year, both perch[65] and pike[68] may undertake short-range migrations from their spawning habitat to deeper feeding areas in winter. However, this does not affect our study as (i) the interactions with stickleback take place during the spawning season, and (ii) our estimate of distance-dependence (Supplementary Fig. 1) used empirical data from tagging and re-captures during the spawning season only. Finally, while the majority of spawning takes place in coastal sites with brackish water, some parts of the Baltic Sea pike and perch populations migrate to adjacent freshwater sites to spawn, and this dependence on freshwater sites can be high locally, in particular for pike[69]. This means that in some places, the availability of spawning habitat may be underestimated.

Based on the created habitat map and our dispersal probability function, we calculated two different measures of connectivity. The first measure was a network-based approach where contiguous patches of habitat (nodes) were linked together based on the least-cost path swimming distance between the centroids of the nodes. We assigned weights to each link based on distance, where weight = 1 − cumulative dispersal probability as described above (see Supplementary Fig. 12 for histogram of link lengths and weights). Only nodes within 10 km or less were assumed to be linked (representing ca 95% of movement; Supplementary Fig. 1). To represent connectivity, we used the weighted sum of connected habitat:

$$connected\ habitat_i = \sum_{1}^{n} weight_{ij} \times area_j \qquad (1)$$

where $i$ is the focal node, and connected habitat is summed across all $n$ connected nodes $j$, weighted by the distance-dependent link weight between $i$ and $j$. Values for individual juvenile sampling points were obtained from the closest node (as the crow flies). See Supplementary Fig. 13 for a histogram of calculated connected area for all juvenile sampling points.

The second measure was instead based on a weighted sum of available habitat around a given juvenile sampling point, calculated

according to the following formula:

$$connected\ habitat_i = \sum_{j=1}^{n} weight_{ij} \times habitat_j \qquad (2)$$

where $i$ is the focal datapoint, $n$ is the total number of raster cells within the maximum dispersal distance of 10 km (ca 95% of all dispersal, see Supplementary Fig. 1), $weight_{ij}$ is the distance-based weight of a given cell $j$ (same approach as for the links above), and $habitat_j$ is a binary value indicating whether cell $j$ contains suitable habitat (1) or not (0). This is then multiplied by 25 (the size of the 5 × 5 m cells) to translate this into areal units. See Supplementary Fig. 14 for a histogram of habitat availability for each juvenile sampling point.

While the network-based approach may more accurately represent the way in which the fish experience and use the habitat, it is more sensitive to the exact configuration of the underlying habitat map, which is why we used two approaches. See Supplementary Fig. 15 for a comparison of values based on the two approaches, and the two cut-offs for wave exposure, and Supplementary Figs. 16–19 for maps.

### Fish consumption by seals and cormorants

Grey seals and great cormorants feed on both the predatory fish (pike and perch) and the stickleback[40,47]. They seem to favour pike and perch, but dietary proportions likely vary over space and time in response to prey availability. Here, we estimate total fish extraction by the seals and cormorants, making no assumptions regarding diet composition. Seal fish consumption was derived from counts at haul-out sites carried out annually by the Swedish Natural History Museum along the Swedish coast (downloaded from the Swedish database for environmental monitoring data; www.sharkweb.smhi.se). The surveys are carried out during the moulting period in May–Early June, with every haul-out site surveyed 2–3 times per year. This is primarily done using aerial surveys, but some counts are also carried out from land or from boats. Here, we used the highest count per year and site based on the logic that a larger proportion of seals was likely in the water during the lower counts. The estimates were further corrected for the fact that ca. 30% of the seals were still expected to be in the water[70]. The data were then translated into a raster of individuals $km^{-2}$ by using a kernel function with a maximum search radius of 60 km (foraging range of Baltic grey seal[71,72]), correcting for the amount of water in each cell. To translate this into the amount of fish extracted from the system, densities were multiplied by 1750 (kg fish eaten per seal and year[73,74]). A similar approach was used for cormorants, making use of two national colony inventories carried out by the Swedish Environmental Protection Agency along the Swedish coast (2006 and 2012), assuming a foraging range of 20 km[75,76] and a consumption rate of 357 kg per nest and breeding season[77]. It is assumed that the cormorants stay in the area only during the breeding season (but see ref. 38). As we only had data from 2006 and 2012, we interpolated values for each cell between these 2 years, and assumed that data from 2006 was representative for all years prior to this (2001–2006) and 2012 representative for all subsequent years (2012–2020). The seal and cormorant layers were then added together to produce annual 1 × 1 km layers of total predation pressure (kg $km^{-2}$ $year^{-1}$; see Supplementary Figs. 20–22), and values for each sampling point were extracted from this layer. Note that the spatial autocorrelation is high (Supplementary Fig. 23a) so that values are also representative for connected areas. All calculations were done in ArcGIS Pro 2.4[61].

### Fish extraction by fishers

Catches from recreational fishers were obtained by distributing catch rate data from national questionnaires (from Statistics Sweden; available for 2006, 2010 and 2013; aggregated at the level of ICES subdivisions) according to spatially explicit population density data[78] combined with information on travel distances for recreational fishing

trips[79] (see ref. [32]). We averaged the maps for all three years (spatial patterns remained similar across years, Supplementary Fig. 24). Yearly data from commercial fisheries were available at the resolution of ICES statistical rectangles from 1999 to 2015. Again, spatial patterns remained similar over time (Supplementary Fig. 25) and so these yearly rasters were also averaged to produce a single raster. The rasters of recreational and commercial catches were added together to produce a final single raster (kg km$^{-2}$ year$^{-1}$ at 1 × 1 km resolution; Supplementary Fig. 26), and values for each sampling point were extracted from this layer. Note that the spatial autocorrelation is high (Supplementary Fig. 23b) so values are also representative of connected areas. The data only covered perch catches, but are also likely to be broadly representative of the spatial distribution of pike catches. All calculations were done in ArcGIS Pro 2.4[61].

## Water temperature

We used sea surface temperature data from the Copernicus Baltic Sea L4 dataset (https://doi.org/10.48670/moi-00156), which is based on reprocessed and interpolated satellite data and available as daily means at a 0.02° × 0.02° resolution (roughly 1–2 km at this latitude)[80]. The agreement with local data is good (Supplementary Fig. 27). For each juvenile sampling data point, we calculated the degree-day sums over a 10 °C threshold (as in ref. [43]), up until the start of the juvenile sampling period towards the end of July, based on data from the closest cell with temperature data (Supplementary Fig. 28). This kind of cumulative temperature metric has been shown to correlate positively with juvenile pike and perch growth and recruitment in the Baltic Sea[21,43].

## Statistical analysis

First, we created a baseline model which modelled the relative dominance of predatory fish (density of pike and perch juveniles/[density of pike, perch and stickleback juveniles])[17] as a function of variables thought to drive the amount of incoming stickleback in an area as described above: density of potential stickleback spawners in the open sea, distance from the open sea, as well as an interaction between the two, and the openness of the area (represented by wave exposure). The model was fit as a generalised linear mixed model assuming a binomial distribution ($N$ = 3491), using the package sdmTMB 0.3.0[81] in R (R 4.2.1[82] was used for all data processing, analysis and visualisation unless explicitly stated). To account for the spatio-temporal structure of the remaining, unexplained variation we fit several models with different random effect structures (combinations of random year effects, spatial random fields, spatio-temporal random fields) and compared these using both AIC and cross-validation (Supplementary Table 8). The spatial random fields are based on a Matérn covariance function, and the distance between observations was based on a mesh that builds on the locations of the data points used in each analysis, a 25 × 25 m resolution water polygon ensuring that the spatial dependencies do not extend across land, as well as parameters controlling the resolution of the mesh. The model including year-specific spatial fields performed the best (Supplementary Table 8) and is therefore presented in the Results.

Then, we explored whether we could explain any of this residual variation using our proposed drivers of resilience: connectivity, predation, fishing, and temperature. We thus refit the models, also including the variables predation pressure, fishing pressure, temperature (degree days) and connectivity (repeating the procedure for the different connectivity metrics separately: wave exposure cut-off of 3.2 vs 3.5, and network-based total connected area vs weighted sum of available habitat), as well as interactions as described in the main text. As described in the Discussion, we had initially also planned to include eutrophication status in the analysis, but, unfortunately, at the spatial scale considered here, the only option is modelled nutrient concentrations with poor alignment with local ground-truthing data. The

prediction errors of the modelled nutrient concentrations also show clear spatial patterns, which could lead to spurious results. They were, therefore, not included in the analysis.

The models were fitted as generalised linear mixed models in the R-package glmmTMB 1.1.7[83], assuming a binomial error distribution. The models also included a random effect of year. Candidate models were compared using AIC (Supplementary Tables 4–6). Effects included in all candidate models with ΔAIC < 4 were considered as supported by the data.

The full analysis was then repeated using predatory fish densities ($N$ = 7415) and stickleback densities ($N$ = 7167) as response variables. Both were modeled using a negative binomial distribution where variance increases quadratically. Note that the sample sizes differ between the different analyses because stickleback density data were not available for some samples, and because relative dominance cannot be calculated when predators and stickleback are both absent (which was the case for more than 50% of the samples).

Since the connectivity measure that showed the best cross-model performance was the second measure (the distance-weighted sum of all available habitat within a 10 km radius) with 3.5 cut-off for wave exposure (Supplementary Table 7), we used this measure for making the plots we present.

In all models, model fit was assessed using the R-package DHARMa 0.4.6[84], and multicollinearity of the explanatory variables was explored using the R-package mctest 1.3.1[85] (see Supplementary Figs. 29–34).

## Reporting summary

Further information on research design is available in the Nature Portfolio Reporting Summary linked to this article.

## Data availability

The majority of the juvenile fish density data are available in the Swedish national database for coastal fish monitoring (slu.se/kul), sea surface temperature data were sourced from the Copernicus Baltic Sea L4 dataset (https://doi.org/10.48670/moi-00156) and seal count data were downloaded from the Swedish database for environmental monitoring data (www.sharkweb.smhi.se). The other datasets used in the study (depth, wave exposure, and landmask rasters, as well as cormorant counts, fisheries data and offshore stickleback densities) were fully or mostly built on data sources that are not publicly available due to, for example, national security concerns or personal authorisation being required before accessing the data. We are happy to discuss inquiries addressed to the corresponding author regarding these datasets. The processed data needed to run the statistical analysis (juvenile fish density data and associated data on drivers for each data point) are available at github.com/agnesolin/stickleback-wave and at doi.org/10.5281/zenodo.10473335[86]. Source data are provided with this paper.

## Code availability

All code can be found at github.com/agnesolin/stickleback-wave and at doi.org/10.5281/zenodo.10473335[86].

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

## Acknowledgements

We are very grateful to everyone involved in the huge effort of collecting data on juvenile fish densities along the Swedish coast, as well as those collecting data on seal and cormorant abundances, fishing activity, and offshore stickleback abundances, without whom this work would not have been possible. We also want to thank Ingo Fetzer for providing access to the SRC server. Finally, we want to thank Jon Norberg, T.J. Clark-Wolf and Gavin Kenny for helpful comments on earlier versions of the manuscript. This work is part of the project "Spatially cascading regime shifts—using network theory to understand and reverse the 'stickleback wave'" (Formas 2019-00310; to J.S.E.) and was finalized within the project "FORCE—Facilitating Ocean Recovery in a Changing climatE" (Formas 2023-00297; to J.S.E.). We have also received in-kind support from Stockholm University (to J.S.E.) and Groningen University (to B.K.E.). The fish juvenile surveys were conducted by a variety of monitoring and research projects, with funding from the Swedish research councils VR and Formas, the Swedish Agency for Marine and Water Management, the Swedish Environmental Protection Agency, the former Swedish Board of Fisheries, the Swedish County administrative boards, the Uppland Foundation, EU Interreg (IIIA and IIIB) and Forsmarks kraftgrupp.

## Author contributions

J.S.E., U.B., Ö.B., G.S. & B.K.E. conceived the study and provided the funding. J.S.E., U.B., Ö.B., G.S., B.K.E. & A.B.O. designed the study. M.E. & R.F. prepared the datasets used in the analysis. A.B.O. processed the

data and conducted the statistical analysis, with J.S.E., U.B., Ö.B., G.S. & B.K.E. providing guidance. A.B.O. wrote the manuscript, with input from all co-authors.

## Funding

## Competing interests

The authors declare no competing interests.
