## [Peer Review File · Nature Communications]

Predation and spatial connectivity interact to shape ecosystem resilience to an ongoing regime shiftREVIEWER COMMENTS

Reviewer #1 (Remarks to the Author):

The manuscript by Olin et al. reports a study on the effect of spatial connectivity on the resilience against progressive dominance of an invading fish species (i.e. stickleback) in the coastal Baltic Sea off Sweden. The paper is based on earlier studies reporting the progressive “stickleback wave” and on an impressive data sets of > 7000 samples. The study is well written and presented, and claims that connectivity and other coastal processes (predation by mammals and birds and fishing on local predators; temperature) have an effect on the resilience against stickleback dominance.

I consider this an interesting study that tries to address ecological theory based on empirical field data which is generally appreciated because similar data sets are rare and experimentation is usually impossible.

The main conclusion of the authors (as given in the title) is that spatial connectivity of the predators reflects the resilience of these against the stickleback invasion. I have a few discussion points that may weaken this bold conclusion.

1) The authors state themselves (lines 160-162) that the resilience measure (probability of predator dominance) can reflect both resistance to and recovery from the stickleback wave. It can hence be a property of the predator populations or the outcome of the invading effect. I agree to the authors that it is difficult/impossible to distinguish between the two. Nevertheless, I think this significantly weakens the outcome of the study, especially as it is used in both ways in the study.

2) The authors found a really strong effect on predator densities and dominance only considering an interaction with fishing pressure. Obviously predation by mammals and birds reduces predator abundance that should also have an effect on connectivity. All other effects were relatively weak, which I think makes it difficult to distinguish if it is just density-dependence (high fishing – reduced predators – reduced dominance) or the weekend connectivity. The latter is generally my main question: is the resilience really due to connectivity or just due to the density of the predators.

3) The authors found also a convincing effect of temperature on predator density and dominance. This effect weakens the general claim of the authors (given by the title) that connectivity is the prime determinant of resilience.

4) In the model comparison the authors found similar explained variance between the baseline predator dominance model (including variables related to the incoming sticklebacks) and the model including connectivity, temperature and their interaction. Differences (R^2 s of 0.23 and 0.38; although I would not call the difference clearly and leaves room for more important factors) were only found for the density model. The latter again makes me wonder if connectivity is really the most important or is it just a density-dependent effect.

In total I would conclude that this is a well done study based on an impressive data set. However, the main conclusion that connectivity is the main driver of resilience to the stickleback wave I find not convincing. First, it could just be the number of predators that determine the resilience (while the number of predators can also be a result of the stickleback invasion). Second, the interaction with the mammals and birds mainly drives the results and temperature is really important for the predator abundance (and their resilience). Hence, although I find the hypothesis and the study setup appealing, I think the results are not supporting the strong claim made by the authors. I nevertheless agree with the authors that taking care of connectivity (through MPAs) is important.

Reviewer #2 (Remarks to the Author):

Key results

Your overview of the key messages of the study, in your own words, highlighting what you find significant or notable. Usually, this can be summarized in a short paragraph.

This study tests the hypothesis that spatial connectivity of species contributes importantly to population resilience and, by extension, ecosystem resilience. The authors leverage a system of fish species (perch/pike versus stickleback) that maintain alternative stable states of the community they inhabit through reciprocal predator-prey interactions with one another. Of multiple potential determinants of these species interactions (e.g., temperature, swell exposure, predation by other species, fishing mortality), individual movement of juveniles among neighboring sites as the mechanism of connectivity is demonstrated to explain spatial variation in the predominance of one group (the “predators”, perch/pike) over the other, and that influence is moderated by other factors (e.g., temperature and predation on the perch/pike). The underlying trigger of the regime shift (ie. what creates shifts in the relative abundance of the species) is unknown (e.g., changes in predation rates on sticklebacks, changes in predation rates or habitat degradation of the perch/pike complex). With the ongoing shifts from the perch/pike to the stickleback-dominated state of these communities, the authors evaluate how spatial connectivity of the perch/pike complex provides that group with resilience (i.e. resistance or recovery) to the stickleback-dominated state. Unfortunately, the authors do not have information on the within-site dynamics of the perch/pike and stickleback populations to determine whether connectivity is maintaining resistance to the influx of sticklebacks or enables recovery (ie. resistance) to the stickleback influx. The significance of the work and its findings is summarized below.

Validity

Your evaluation of the validity and robustness of the data interpretation and conclusions. If you feel there are flaws that prohibit the manuscript’s publication, please describe them in detail.

The authors draw from a truly impressive collection of past studies and their own analyses to evaluate the relative contribution of population connectivity to the resistance or resilience of these populations and communities. I am entirely unfamiliar with the system, but the authors leverage a system that very nicely exemplifies changing alternative states. While there is much known of the system, there are two elements that remain unknown and the authors point these deficiencies out. First, we don’t know what causes the shifts in alternative states (species dominance) and, secondly, whether connectivity is conferring resistance (doesn’t change) or resilience (changes and recovers) to the perturbation. Readers are certain to have concerns with these deficiencies, but as the authors state and I agree, the demonstration of connectivity in preventing the system to shift is of great importance even in the absence of this information. They need to be straightforward of this in the Discussion section and emphasize the importance of filling both knowledge gaps.

I do have a couple of fundamental concerns with the study that I feel the authors need to clarify, especially for those of us who are unfamiliar with this system.

Given the geographic area of the study region across the extensive island archipelago, and the stickleback wave from “offshore” to the mainland, it is very easy for readers (especially marine people like myself) to envision that these species occur throughout coastal habitats. Especially because the emphasis on embayments is “their preferred spawning habitat”, suggesting that they migrate to these from other habitats from alternative habitat, as many marine species do. But the reality, as I understand it, is that (1) they only occur in these embayments, (2) those embayments are distributed throughout the archipelago (offshore to mainland), and (3) the connectivity is among the embayments. This is depicted in Figure 4. If this interpretation is correct, then please be sure this is crystal clear in the description of the system in the Introduction and Methods.

Secondly, I need a better explanation as to why “we considered connectivity as the amount of accessible neighbouring predator spawning habitat. Both pike and perch have a particular preference for spawning in shallow and sheltered bays^{59,62}” (Line 399) and how this aligns with the life history of these species. These species produce larvae that are pelagically dispersed, however that mechanism of dispersal and the resulting patterns of juvenile density are not accounted for in the models. Why is that? Are there not two forms of connectivity? (1) Combined larval and juvenile connectivity, which first involves (i) distance from spawning habitat to suitable settlement habitat (determined by larval dispersal distance), then (ii) area of suitable habitat across which settled juveniles transit (determined by Suppl. Figure 1) from where they settle to the nearest spawning habitat. (2) Adult connectivity (using juvenile movement distance as a proxy), by which adults move from one embayment to another as a function of proximity between embayments and suitable habitat for adults to transit from one embayment to another. For (1), isn't the resulting connectivity and density distribution of juveniles a function of larval production at the spawning site, larval dispersal distance and the amount of available settlement habitat (= larval connectivity), and the availability (area) of habitat for settled juveniles to subsequently move to adult spawning habitat (juvenile connectivity)? If, in fact, this is what is being modelled (rather than juveniles emigrating from spawning habitat, which doesn't make sense for species with pelagic larvae), I think this needs to be made clearer. Including description of larval duration and dispersal distances. Maybe the approach of estimating connectivity covers both (1) and (2), but it should be clarified.

This is not a concern, but just a comment. It is quite interesting that the impact of the seals and cormorants on the perch and pike doesn't translate to increased density of sticklebacks.

Significance

Your view on the potential significance of the conclusions for the field and related fields. If you think that other findings in the published literature compromise the manuscript's significance, please provide relevant references.

With ever-growing perturbations to populations and ecosystems, especially in the face of global climate change, a crucial role of ecology is to better understand what attributes of populations, communities, ecosystems enable these systems to be resistant or resilient to these perturbations that otherwise can result in changes to alternative states. As such, studies that identify system attributes or mechanisms that confer resistance or resilience are very important, both for advancing ecology and informing conservation management and policy (e.g., protecting those traits). Of the many systems attributes that might confer resistance or resilience, ecological connectivity, in this case population connectivity, is one of growing interest and recognition. Hence the interest in “ecological corridors” to facilitate that

connectivity. Yet, as the authors mention, while there are lots of theory and models, there are few good empirical examples to demonstrate the role of organismal connectivity for enhancing system resistance or resilience. The great significance of this study is the conclusion that the study demonstrates empirically how population connectivity facilitates not just population, but ecosystem resistance or resilience. Thus, with the caveats mentioned above, I agree with the authors of the great significance of the study.

If the authors can clarify the mechanisms of connectivity that they do and don't model and how, then I think the manuscript will be a great contribution.

Data and methodology

Your assessment of the validity of the approach, the quality of the data, and the quality of presentation. We ask reviewers to assess all data, including those provided as supplementary information. If any aspect of the data is outside the scope of your expertise, please note this in your report or in the comments to the editor. We may, on a case-by-case basis, ask reviewers to check code provided by the authors (see this Nature editorial for more information).

With the caveat of the comments about different forms of connectivity, the authors' synthesis of such varied sources of information is really impressive. I do believe the methods as described and the data they have made available, including that analyses in the following section, are sound and reproducible.

Analytical approach

Your assessment of the strength of the analytical approach, including the validity and comprehensiveness of any statistical tests. If any aspect of the analytical approach is outside the scope of your expertise, please note this in your report or in the comments to the editor.

I have no concerns with how the authors analyzed the data to test their hypotheses, including how they estimated connectivity and the spatial patterns of other ecological and environmental variables. The relationships from the GAMs and the spatial patterns presented in the Supplemental Information were extremely useful.

Suggested improvements

Your suggestions for additional experiments or data that could help strengthen the work and make it suitable for publication in the journal. Suggestions should be limited to the present scope of the manuscript; that is, they should only include what can be reasonably addressed in a revision and exclude what would significantly change the scope of the work. The editor will assess all the suggestions received and provide additional guidance to the authors.

I'm confident that no data are available to distinguish resistance and resilience (within population dynamics of the perch/pike versus sticklebacks) or the driver of shifting states, or the authors would have conducted those analyses. Everyone is now dying to know whether eutrophication of some other anthropogenic factor is altering the balance of predominance in the embayments. But is that happening on the more isolated embayments offshore where the system is actively shifting?

There is one other aspect of this study that would be a very valuable contribution. Not to fall too deeply here into metapopulation and metacommunity theory, but it would be interesting (for their next paper?)

to know if persistence of the perch/pike predominance is maintained by asynchronous source-sink dynamics. Are declining spawning populations saved by connectivity with source populations before they switch?

A more relevant metapopulation question for any connectivity involving adults moving among embayments is whether emigration is density-dependent. Is density dependence the driver of migrants to seek out alternative embayments and rescue those populations?

There is a very important conservation implication of this work that could be better emphasized in the Discussion or overall Conclusion. It is a fundamental nature of these open marine systems and rationale for networks of marine protected areas that are ecologically networked by the dispersal of organisms. These results imply that managing for the quality of the spawning habitat in one embayment such that it favors the predatory fishes at the cost to sticklebacks has consequences for these embayment ecosystems well beyond the single managed embayment.

I would like to see my concerns raised in the “Validity” section addressed before publication.

Clarity and context

Your view on the clarity and accessibility of the text, and whether the results have been provided with sufficient context and consideration of previous work. Note that we are not asking for you to comment on language issues such as spelling or grammatical mistakes.

This is an opportunity to raise my perspective on the author’s use of the terms resilience “against” (lines 31, 42, 230, 289) or “towards” a shift (lines 53, 108, 111, 158, 232, 237). How is it that a system can be resilient both “against” and “towards” a shift? I find the use of both of these confusing. More importantly, I don’t like either of them. I am accustomed to systems simply being resilient to or from perturbation or change. They change and recover “from” a perturbation (or change) and are therefore resilient “to” that that perturbation (or change). Perhaps semantics, but I wanted to share that.

References

Your view on whether the manuscript references previous literature appropriately.

These are fine.

Below you can find point-by-point responses to the reviewer comments, where responses are marked in green and line numbers refer to those in the new version of the manuscript, with changes accepted.

REVIEWER COMMENTS

Reviewer #1 (Remarks to the Author):

The manuscript by Olin et al. reports a study on the effect of spatial connectivity on the resilience against progressive dominance of an invading fish species (i.e. stickleback) in the coastal Baltic Sea off Sweden. The paper is based on earlier studies reporting the progressive “stickleback wave” and on an impressive data sets of > 7000 samples. The study is well written and presented, and claims that connectivity and other coastal processes (predation by mammals and birds and fishing on local predators; temperature) have an effect on the resilience against stickleback dominance.

I consider this an interesting study that tries to address ecological theory based on empirical field data which is generally appreciated because similar data sets are rare and experimentation is usually impossible.

The main conclusion of the authors (as given in the title) is that spatial connectivity of the predators reflects the resilience of these against the stickleback invasion. I have a few discussion points that may weaken this bold conclusion.

1) The authors state themselves (lines 160-162) that the resilience measure (probability of predator dominance) can reflect both resistance to and recovery from the stickleback wave. It can hence be a property of the predator populations or the outcome of the invading effect. I agree to the authors that it is difficult/impossible to distinguish between the two. Nevertheless, I think this significantly weakens the outcome of the study, especially as it is used in both ways in the study.

Response: We fully agree that further understanding of the role of recovery vs resistance is important, and we have added a sentence to this effect (L350-351). As the reviewer points out, it is unfortunately difficult, if not impossible, to adequately distinguish between resistance and recovery using the available data (see L. 177-185). We do, however, have a few locations with repeated sampling, which could at least provide a starting point for addressing this issue, even though we do not believe the data are sufficient for a full analysis that can be included in the manuscript. Below, we have included time series plots of relative predator dominance (RPD) in all locations with at least 8 years of data (with a minimum of 5 detonations in each year). Each location generally corresponds to a bay, but note that delimitation into bays is not always straightforward so that some areas may represent a longer stretch of coast. Each year is colour-coded into predator-dominated (green, $RPD \leq 0.1$), stickleback-dominated (red, $RPD \geq 0.9$), or mixed (grey). Some locations are consistently predator-dominated (e.g. Sundbymaren, Fjällviksviken), others are consistently stickleback-dominated (e.g.

Ängsösundet, Utöfladen), while some seem to be undergoing/have undergone a transition to stickleback-domination (e.g. Flacket, Forsmark) and others flicker between the two states (e.g. Mjölkviken, Byviken). The data suggest that a flip from predator-domination to stickleback-domination is a longer process, which may involve a period of flickering, as is commonly observed in regime shifts and is considered as an “early-warning signal” (Scheffer et al. 2009). Likely, a mix of processes may be involved in this flickering period, which alter the dominance dynamics in a given year. For example, the amount of incoming stickleback will vary from year to year as a result of e.g. overwinter survival offshore (see interannual variation in Fig 2a-c in Olin et al. 2022), while predator juvenile recruitment and growth rate (and thus their vulnerability to stickleback predation) may vary as a result of e.g. temperature (as found in this study). That a location can “bounce back” to predator-domination can be interpreted at least as potential for recovery (even if it does not represent recovery from a stable flip), while the years of intermediate dominance in the course of an ongoing flip (see Flacket, Forsmark) may hint at resistance. No time series show a more stable flip back to predator-dominance, however, suggesting that we are not seeing any long-term recovery at this moment in time.

As such, these time series (albeit short) provide us with some clues regarding the mechanisms of resistance and recovery. However, to fully distinguish between resistance and recovery using these time series, we would need to have good estimates of the stressor (i.e. the amount of incoming stickleback). While we do estimate this in the study, this is a relatively coarse estimate that relies on applying it to a large dataset to elucidate any effect (3491–7415 datapoints, depending on the response variable). As such, we do not believe that it is precise enough to make a formal statistical analysis meaningful given the relatively small dataset (N = 25 time series of varying length). Even if we would be able to adequately account for the level of the stressor, this would leave little statistical power to examine the role of connectivity or any other variable. However, if these plots are a helpful complement for understanding the underlying mechanisms, we would be happy to add them to the supplementary materials, together with some qualitative interpretation in the Discussion.

Finally, even if these two mechanisms cannot be disentangled, resistance and recovery are both important components of resilience (L44-46) and we believe that the study still provides important information on the functioning and resilience of the system (L 182-185).

Olin, A. B., Olsson, J., Eklöf, J. S., Eriksson, B. K., Kaljuste, O., Briekmane, L., & Bergström, U. (2022). Increases of opportunistic species in response to ecosystem change: the case of the Baltic Sea three-spined stickleback. *ICES Journal of Marine Science*, 79, 1419–1434.

Scheffer, M., Bascompte, J., Brock, W.A., Brovkin, V., Carpenter, S.R., Dakos, V., Held, H., Van Nes, E.H., Rietkerk, M. and Sugihara, G. (2009). Early-warning signals for critical transitions. *Nature*, 461, 53–59.

2) The authors found a really strong effect on predator densities and dominance only considering an interaction with fishing pressure. Obviously predation by mammals and birds reduces predator abundance that should also have an effect on connectivity. All other effects were relatively weak, which I think makes it difficult to distinguish if it is just density-dependence (high fishing – reduced predators – reduced dominance) or the weekend connectivity. The latter is generally my main question: is the resilience really due to connectivity or just due to the density of the predators.

Response: Since there was only support for an interaction between connectivity and predation pressure from seals and cormorants, we assume that in this comment “fishing pressure” and “high fishing” refer to predation by seals and cormorants (or, in theory, any driver that results in increased predator mortality).

We fully agree that predation by mammals and birds is likely to affect the abundance of the predators and, by extension, the realised connectivity (i.e., the amount of fish actually moving between habitat patches). This is what we try to capture by including an interaction between predation pressure (and also fishing pressure) and our connectivity measure (which in itself captures only physical/potential connectivity, i.e., the presence of habitat and dispersal routes, which means little for pike and perch densities if there are no pike and perch to move between the habitat patches). We have now tried to clarify our reasoning (L129-138).

Our thinking is that if only an effect of predation (and/or fishing) was at play, the effect of connectivity (and its interaction with predation/fishing) would have dropped out of the models,

which would allow us to distinguish between an effect of connectivity and high predation/fishing – reduced predators – reduced dominance.

Finally, we fully agree that the density of the (adult) predators is really what makes up the resilience here (as shown by previous work; e.g. Eklöf et al. 2020). We have tried to make this clearer in the manuscript (L110-115, L132). The idea is that connectivity is one out of several drivers, all of which we unfortunately cannot capture here due to a lack of data, that will affect this density of adult predators. We also do not believe that connectivity is the most important of these drivers (either in theory or in our findings), although we realise it may have come across as this in our writing. We have rephrased our wording to make it clearer that we do not consider connectivity to be the main driver (L383-384). The focus on connectivity instead reflects our belief that this finding is the most novel and important contribution of the study.

Eklöf, J. S. *et al.* A spatial regime shift from predator to prey dominance in a large coastal ecosystem. *Commun. Biol.* **3**, 459 (2020).

3) The authors found also an convincing effect of temperature on predator density and dominance. This effect weakens the general claim of the authors (given by the title) that connectivity is the prime determinant of resilience.

Response: Again, we realise that it may have come across as if we consider connectivity to be the main driver. However, we do not believe that the title suggests that connectivity is the main driver of resilience, only that it increases resilience, and we therefore wish to leave the title as it is. We have also tried to make it clearer how much variation is explained by the different drivers by calculating R^2 -values without each driver (L245-247; L272-273; L393-297; L321-323).

4) In the model comparison the authors found similar explained variance between the baseline predator dominance model (including variables related to the incoming sticklebacks) and the model including connectivity, temperature and their interaction. Differences (R^2 s of 0.23 and 0.38; although I would not call the difference clearly and leaves room for more important factors) were only found for the density model. The latter again makes me wonder if connectivity is really the most important or is it just a density-dependent effect.

Response: 0.38 is still 65% higher than 0.23, which we consider quite a substantial difference. We agree, however, that this still leaves a lot of unexplained variation, which we have added a comment on (L373-374). We also agree that the difference between the full model and the model only including variables related to incoming stickleback was small for the relative dominance (L334). Again, we have also clarified that we do not consider connectivity the most important driver (L383-384).

In total I would conclude that this is a well done study based on an impressive data set. However, the main conclusion that connectivity is the main driver of resilience to the stickleback wave I find not convincing.

Response: As stated above, we do not consider connectivity as the main driver of resilience, and we hope that we have now clarified this through the changes detailed above. While we still argue that the result regarding the relationship between connectivity and resilience is the most scientifically interesting and novel finding of our study, we have now rephrased aspects of the text to emphasise that this does not necessarily mean that it is the most influential driver.

First, it could just be the number of predators that determine the resilience (while the number of predators can also be a result of the stickleback invasion). Second, the interaction with the mammals and birds mainly drives the results and temperature is really important for the predator abundance (and their resilience). Hence, although I find the hypothesis and the study setup appealing, I think the results are not supporting the strong claim made by the authors. I nevertheless agree with the authors that taking care of connectivity (through MPAs) is important.

Response: As correctly pointed out, the incoming stickleback will also affect the density of predators. However, in the model we have controlled for this to the best of our ability by including the density of offshore stickleback, their migration distance and the openness of the area (L147-165; L423-439). We agree that it is important to not exaggerate the role of connectivity, but we also believe that the identified effect is an important finding, and have therefore clarified the relative importance of connectivity by highlighting how much variation it helps explain in relation to other drivers (L245-247; L272-273; L393-297; L321-323).

Reviewer #2 (Remarks to the Author):

Key results

Your overview of the key messages of the study, in your own words, highlighting what you find significant or notable. Usually, this can be summarized in a short paragraph.

This study tests the hypothesis that spatial connectivity of species contributes importantly to population resilience and, by extension, ecosystem resilience. The authors leverage a system of fish species (perch/pike versus stickleback) that maintain alternative stable states of the community they inhabit through reciprocal predator-prey interactions with one another. Of multiple potential determinants of these species interactions (e.g., temperature, swell exposure, predation by other species, fishing mortality), individual movement of juveniles among neighboring sites as the mechanism of connectivity is demonstrated to explain spatial variation in the predominance of one group (the “predators”, perch/pike) over the other, and that influence is moderated by other factors (e.g., temperature and predation on the perch/pike). The underlying trigger of the regime shift (ie. what creates shifts in the relative abundance of the species) is unknown (e.g., changes in predation rates on sticklebacks, changes in predation rates or habitat degradation of the perch/pike complex). With the ongoing shifts from the perch/pike to the stickleback-dominated state of these communities, the authors

evaluate how spatial connectivity of the perch/pike complex provides that group with resilience (i.e. resistance or recovery) to the stickleback-dominated state. Unfortunately, the authors do not have information on the within-site dynamics of the perch/pike and stickleback populations to determine whether connectivity is maintaining resistance to the influx of sticklebacks or enables recovery (ie. resistance) to the stickleback influx. The significance of the work and its findings is summarized below.

Validity

Your evaluation of the validity and robustness of the data interpretation and conclusions. If you feel there are flaws that prohibit the manuscript's publication, please describe them in detail.

The authors draw from a truly impressive collection of past studies and their own analyses to evaluate the relative contribution of population connectivity to the resistance or resilience of these populations and communities. I am entirely unfamiliar with the system, but the authors leverage a system that very nicely exemplifies changing alternative states. While there is much known of the system, there are two elements that remain unknown and the authors point these deficiencies out. First, we don't know what causes the shifts in alternative states (species dominance) and, secondly, whether connectivity is conferring resistance (doesn't change) or resilience (changes and recovers) to the perturbation. Readers are certain to have concerns with these deficiencies, but as the authors state and I agree, the demonstration of connectivity in preventing the system to shift is of great importance even in the absence of this information. They need to be straightforward of this in the Discussion section and emphasize the importance of filling both knowledge gaps.

Response: Good points. We have added some more discussion on resistance vs recovery in both the Results (L177-188) and the Discussion (L345-352), including a statement on the importance of filling this knowledge gap (L350-351). The ultimate causes of the shift are likely to involve mainly drivers of the stickleback increase, but also drivers of predator population declines, and we have now emphasised the importance of trying to understand both on L336-337 and L.373-374.

I do have a couple of fundamental concerns with the study that I feel the authors need to clarify, especially for those of us who are unfamiliar with this system. Given the geographic area of the study region across the extensive island archipelago, and the stickleback wave from "offshore" to the mainland, it is very easy for readers (especially marine people like myself) to envision that these species occur throughout coastal habitats. Especially because the emphasis on embayments is "their preferred spawning habitat", suggesting that they migrate to these from other habitats from alternative habitat, as many marine species do. But the reality, as I understand it, is that (1) they only occur in these embayments, (2) those embayments are distributed throughout the archipelago (offshore to mainland), and (3) the connectivity is among the embayments. This is depicted in Figure 4. If this interpretation is correct, then please be sure this is crystal clear in the description of the system in the Introduction and Methods.

Response: This description is mostly correct, although the pike and perch do indeed undertake some short-range migration. We fully agree that it is important for the reader to have a clear understanding of the system and have now amended and clarified the text to make it easier to follow:

- (1) The perch and pike, which are of freshwater origin, are closely tied to sheltered, shallow coastal habitat, often in embayments but not necessarily (we have clarified this in the text, L449-451). During winter, they undertake short migrations to feeding areas, but still remain close to the coast (Siddika & Lehtonen 2004; Karås & Lehtonen 1993; added on L467-469). In this study we are only interested in interactions during the spawning season; the time of the year when the main interactions with stickleback occur (L442-444). This means that the distribution of the perch and pike in winter does not matter much in this context (L469-472). Further, some fraction of the populations also spawn in freshwater habitat (Engstedt et al. 2015; Wastie 2014), and the contribution from freshwater spawning likely varies over space. We have clarified this in the text (L472-476), and also highlight the importance of understanding the role of access to freshwater habitat (L371-372).
- (2) This shared spawning habitat is indeed distributed throughout the archipelago, which we have now clarified (L61).
- (3) Yes, connectivity is indeed between these embayments, or more correctly, between patches of spawning habitat (as, again, this is often but not always in an embayment). We have added a sentence on L131 that hopefully makes it clear enough even to readers who do not make it to the Methods section.

Karås, P., Lehtonen, H. (1993). Patterns of movement and migration of pike (*Esox Lucius* L.) in the Baltic Sea. *Nordic Journal of Freshwater Research*, 68, 1159–1172.

Engstedt, O., Stenroth, P., Larsson, P., Ljunggren, L., Elfman, M. (2015). Assessment of natal origin of pike (*Esox Lucius*) in the Baltic Sea using Sr:Ca in otoliths. *Environmental Biology of Fishers*, 89, 547–555.

Siddika, M. K. & Lehtonen, H. (2004). Spawning migration of perch, (*Perca fluviatilis* L.), in different coastal waters in the Baltic Sea. *IIFET 2004 Jpn. Proc.* 11.

Wastie, J. (2014). Assessing the importance of freshwater tributary systems for the recruitment of Eurasian perch (*Perca fluviatilis*) in Baltic Sea coastal ecosystems. Masters thesis, Swedish University of Agricultural Sciences, Öregrund, Sweden.

Secondly, I need a better explanation as to why “we considered connectivity as the amount of accessible neighbouring predator spawning habitat. Both pike and perch have a particular preference for spawning in shallow and sheltered bays^{59,62}” (Line 399) and how this aligns with the life history of these species. These species produce larvae that are pelagically dispersed, however that mechanism of dispersal and the resulting patterns of juvenile density are not accounted for in the models. Why is that? Are there not two forms of connectivity? (1) Combined larval and juvenile connectivity, which first involves (i) distance from spawning habitat to suitable settlement habitat (determined by larval dispersal distance), then (ii) area of suitable habitat across which settled juveniles transit (determined by Suppl. Figure 1) from where they settle to the nearest spawning habitat. (2) Adult connectivity (using juvenile movement distance as a proxy), by which adults move from one embayment to another as a

function of proximity between embayments and suitable habitat for adults to transit from one embayment to another. For (1), isn't the resulting connectivity and density distribution of juveniles a function of larval production at the spawning site, larval dispersal distance and the amount of available settlement habitat (= larval connectivity), and the availability (area) of habitat for settled juveniles to subsequently move to adult spawning habitat (juvenile connectivity)? If, in fact, this is what is being modelled (rather than juveniles emigrating from spawning habitat, which doesn't make sense for species with pelagic larvae), I think this needs to be made clearer. Including description of larval duration and dispersal distances. Maybe the approach of estimating connectivity covers both (1) and (2), but it should be clarified.

Response: This is a very good point, and we agree that we had not explained this clearly enough in the manuscript. The larvae of pike and perch move very short distances (e.g. perch larval dispersal estimated to 0.1-2 km; Berkström et al. 2022; over a duration of ca. 2 months, depending on temperature; Karås 1996), before metamorphosing and returning to the littoral zone. We have now clarified this in the manuscript (L463-465). For people familiar with marine fishes, the short dispersal distances of perch and pike larvae may come as a surprise, but is an adaptation to the harsh conditions of the Baltic Sea, where these larvae are dependent on the warm water only found in enclosed bays in spring. Unfortunately, we do not have any data to account for larval dispersal explicitly (e.g. any estimate of probability of larval dispersal as a function of distance). However, considering the highly localised larval dispersal, we believe that it is unlikely to affect our dispersal probability-distance function much, which now represents *adult* movement only, but is assumed to also be representative for juvenile dispersal (Fig. S1). Further support for the idea that the larval (or juvenile) dispersal does not result in greater distances of connectivity than assumed here comes from genetic studies suggesting genetic isolation at very short distances (a few kilometres; Bergek & Björklund 2009; Bergek et al. 2010; see L465-466). We agree, however, that the dynamics of dispersal during the first year of life is important, and that future studies on distances between the location in which an individual is spawned, and the location in which it later spawns itself, would be highly useful (see additions on L274-276). Our current metrics also do not account for the fact that the characteristics of the intervening habitat between spawning habitat patches may affect the propensity to disperse and the route taken, where further studies would allow for better estimates of connectivity (see L273-274).

Bergek, S. & Björklund, M. (2009). Genetic and morphological divergence reveals local subdivision of perch (*Perca fluviatilis* L.), *Biological Journal of the Linnean Society*, 96, 746–758

Bergek, S., Sundblad, G., & Björklund, M. (2010). Population differentiation in perch *Perca fluviatilis*: environmental effects on gene flow? *Journal of Fish Biology*, 76, 1159–1172.

Berkström, C., Wennerström, L. & Bergström, U. (2021). Ecological connectivity of the marine protected area network in the Baltic Sea, Kattegat and Skagerrak: Current knowledge and management needs. *Ambio*. 51, 1485–1503.

Karås, P. (1996) Basic abiotic conditions for production of perch (*Perca fluviatilis* L.) young-of-the-year in the Gulf of Bothnia. *Annales Zoologici Fennici*, 33, 371–381.

This is not a concern, but just a comment. It is quite interesting that the impact of the seals and cormorants on the perch and pike doesn't translate to increased density of sticklebacks.

Response: We agree that this is interesting. As described in the text, seals and cormorants also feed on stickleback (L289) and as seals and cormorants are opportunistic feeders, it makes sense that they would crop down the population of stickleback in bays where the stickleback is dominant and abundant. In the future, we are hoping to study the dynamics over time, to see if we can observe that an increase in seals and cormorants in a given location are followed by less perch and pike, and more stickleback, to explore the possibility of a five-level trophic cascade all the way to algae (see L297-300).

Significance

Your view on the potential significance of the conclusions for the field and related fields. If you think that other findings in the published literature compromise the manuscript's significance, please provide relevant references.

With ever-growing perturbations to populations and ecosystems, especially in the face of global climate change, a crucial role of ecology is to better understand what attributes of populations, communities, ecosystems enable these systems to be resistant or resilient to these perturbations that otherwise can result in changes to alternative states. As such, studies that identify system attributes or mechanisms that confer resistance or resilience are very important, both for advancing ecology and informing conservation management and policy (e.g., protecting those traits). Of the many systems attributes that might confer resistance or resilience, ecological connectivity, in this case population connectivity, is one of growing interest and recognition. Hence the interest in "ecological corridors" to facilitate that connectivity. Yet, as the authors mention, while there are lots of theory and models, there are few good empirical examples to demonstrate the role of organismal connectivity for enhancing system resistance or resilience. The great significance of this study is the conclusion that the study demonstrates empirically how population connectivity facilitates not just population, but ecosystem resistance or resilience. Thus, with the caveats mentioned above, I agree with the authors of the great significance of the study.

If the authors can clarify the mechanisms of connectivity that they do and don't model and how, then I think the manuscript will be a great contribution.

Response: We are happy to hear that we are on the same page regarding the general interest in how connectivity may contribute to ecosystem resilience. We hope that we have managed to better describe what is included in our measure of connectivity.

Data and methodology

Your assessment of the validity of the approach, the quality of the data, and the quality of presentation. We ask reviewers to assess all data, including those provided as supplementary information. If any aspect of the data is outside the scope of your expertise, please note this in your report or in the comments to the editor. We may, on a case-by-case basis, ask reviewers to check code provided by the authors (see this Nature editorial for more information).

With the caveat of the comments about different forms of connectivity, the authors' synthesis of such varied sources of information is really impressive. I do believe the methods as described and the data they have made available, including that analyses in the following section, are sound and reproducible.

Response: Thank you.

Analytical approach

Your assessment of the strength of the analytical approach, including the validity and comprehensiveness of any statistical tests. If any aspect of the analytical approach is outside the scope of your expertise, please note this in your report or in the comments to the editor.

I have no concerns with how the authors analyzed the data to test their hypotheses, including how they estimated connectivity and the spatial patterns of other ecological and environmental variables. The relationships from the GAMs and the spatial patterns presented in the Supplemental Information were extremely useful.

Response: Thank you.

Suggested improvements

Your suggestions for additional experiments or data that could help strengthen the work and make it suitable for publication in the journal. Suggestions should be limited to the present scope of the manuscript; that is, they should only include what can be reasonably addressed in a revision and exclude what would significantly change the scope of the work. The editor will assess all the suggestions received and provide additional guidance to the authors.

I'm confident that no data are available to distinguish resistance and resilience (within population dynamics of the perch/pike versus sticklebacks) or the driver of shifting states, or the authors would have conducted those analyses. Everyone is now dying to know whether eutrophication of some other anthropogenic factor is altering the balance of predominance in the embayments. But is that happening on the more isolated embayments offshore where the system is actively shifting?

Response: While there are some time series on juvenile densities available (see response to reviewer #1 above), the lack of site-specific data on the stressor (i.e. the amount of incoming stickleback) prevents us from making full use of these as a way of understanding the internal dynamics, or the role of resistance vs recovery. Further, these time series of juvenile densities should ideally also be complemented by time series of the adults to fully understand the changing dynamics. However, if presenting the time series in the supplementary materials would help in providing a better understanding of the system, we would be happy to include them.

As mentioned in the text (L374-377; L584-589), eutrophication would have been a very interesting variable to include, but unfortunately there are no high-resolution data available for the study area. We do include fishing, which is an anthropogenic variable that may also be

correlated with overall disturbance (coastal construction, boating etc), which is also likely to affect the dominance balance (L308-310). In the more isolated embayments in the outermost archipelago, the anthropogenic pressure is likely lower but at the same time, the amount of incoming stickleback is large and sheltered bays with good temperature development are harder to find. Consequently, these bays were also the first to shift (L70-71).

There is one other aspect of this study that would be a very valuable contribution. Not to fall too deeply here into metapopulation and metacommunity theory, but it would be interesting (for their next paper?) to know if persistence of the perch/pike predominance is maintained by asynchronous source-sink dynamics. Are declining spawning populations saved by connectivity with source populations before they switch? A more relevant metapopulation question for any connectivity involving adults moving among embayments is whether emigration is density-dependent. Is density dependence the driver of migrants to seek out alternative embayments and rescue those populations?

Response: These are very good points and we agree that it would be worth investigating directional movement. The effects of density-dependent dispersal could potentially have contrasting effects on resilience: either resulting in spread out (“smeared out”), low-density populations that cannot keep the stickleback at bay (similar to what we describe on L264-269), or the density-dependent dispersal forces the predators to disperse to alternative spawning bays and “rescue” these populations, as you suggest. We have added a sentence on directional dispersal as an important area of future research (L276-279).

There is a very important conservation implication of this work that could be better emphasized in the Discussion or overall Conclusion. It is a fundamental nature of these open marine systems and rationale for networks of marine protected areas that are ecologically networked by the dispersal of organisms. These results imply that managing for the quality of the spawning habitat in one embayment such that it favors the predatory fishes at the cost to sticklebacks has consequences for these embayment ecosystems well beyond the single managed embayment.

Response: We agree that this is an interesting implication of our work, and have modified a sentence in the conclusion paragraph to emphasise this point (L389-391).

I would like to see my concerns raised in the “Validity” section addressed before publication.

Response: Again, we hope we have managed to make our description of the study system and of our connectivity metric sufficiently clear.

Clarity and context

Your view on the clarity and accessibility of the text, and whether the results have been provided with sufficient context and consideration of previous work. Note that we are not asking for you to comment on language issues such as spelling or grammatical mistakes.

This is an opportunity to raise my perspective on the author's use of the terms resilience "against" (lines 31, 42, 230, 289) or "towards" a shift (lines 53, 108, 111, 158, 232, 237). How is it that a system can be resilient both "against" and "towards" a shift? I find the use of both of these confusing. More importantly, I don't like either of them. I am accustomed to systems simply being resilient to or from perturbation or change. They change and recover "from" a perturbation (or change) and are therefore resilient "to" that that perturbation (or change). Perhaps semantics, but I wanted to share that.

Response: This is an interesting and important point. We have used to/against/towards as practically interchangeable in this context but we now realise that this is not how it may be interpreted by everyone. We agree with the view of a system being resilient "to" a perturbation/change and have adjusted our wording throughout to avoid any confusion.

References

Your view on whether the manuscript references previous literature appropriately.

These are fine.

Response: Thank you.

REVIEWERS' COMMENTS

Reviewer #2 (Remarks to the Author):

Comments on revised ms:

General comments:

As usual, the revised manuscript is much improved. Much of the improvement is based on the added text that directly addresses the concerns of both reviewers. I find the revision more compelling and a clearer characterization of the system, the results, and the author's interpretations.

The revised manuscript, especially the added lines 265-268 and lines 318-321(!), makes clear that connectivity is only one important contributor to predator resilience. While I very much appreciate the emphasis on the contribution of connectivity to system resilience, I suggest the authors re-think the title "Spatial connectivity increases ecosystem resilience to an ongoing regime shift" to be more inclusive "Predation, spatial connectivity, and the local environment interact to increase ecosystem resilience to an ongoing regime shift". I say this because (1) it still reinforces the importance of spatial connectivity as a driver of ecosystem resilience, but (2) goes a step further to indicate that the connectivity effect interacts with other drivers (predation, temperature). It's an even greater advance than identifying the role of connectivity alone. Those looking for evidence of the importance of species interactions (predator-prey), spatial connectivity, and environmental conditions (temperature) and their interactions(!) will be drawn to the value of this article.

I have the remaining recommendations:

I do think it is necessary to include the time series that the authors provided the reviewers with as supplementary information in the revised manuscript! What I find especially interesting, depending on the exposure of these sites to stickleback recruitment, is the 5 of 25 sites that persisted as predator-dominated, suggesting resistance to invasion. If these 5 sites were subjected to stickleback recruitment, it would be strong evidence for resistance while the other sites exhibit greater vulnerability to transitioning to the stickleback dominated state. And the question is whether these 5 sites experienced greater predator connectivity than the others.

Line 144: Delete comma after "predation/fishing"

Line 205: "effect of on the absolute densities of predator juveniles..." Effect of connectivity?

Line 270 (first paragraph of Discussion) While I appreciate the emphasis on connectivity to highlight its importance to resilience, this first sentence summarizing the drivers of resilience doesn't mention what the authors identified as a crucial driver, predation. I would restate this to say that they found that interactions of predation, connectivity and the environment (temperature) all contributed critically to estimates of resistance/resilience of the predator-dominated state of the ecosystem. Then the rest of this paragraph describes those interactions.

Line 280: In this section, simply make clear here that the vehicle of connectivity is the adult stage. "The movement of adult predatory fish between spawning areas allows for redistribution and recolonisation." and "high predation pressure from top predators, most likely because there were too few adult predators to redistribute." The revised Methods that includes information on larval dispersal is invaluable for clarifying why adult movement is focused on here. But it is good to clarify it in this section, rather than the reader having to read the Methods to understand that adult movement is the basis of connectivity here and why.

Line 375: This new text is extremely valuable for teasing apart the relative influences of resistance and recovery (how I define resilience) as they contribute to resilience or persistence of the predator-dominated state. As you know, there is an extensive literature on the importance of resistance on

ecosystem invasibility and the mechanisms of resistance (e.g., species diversity and identity) versus capacity for recovery (e.g., changes in species recruitment rates) can be fundamentally different. This paragraph helps to address this.

Line 384: Among the many potential management implications identified, I didn't see one that truly reflects the role of connectivity, which is to ensure that connectivity corridors are maintained. If human activities can impact the quality of habitat required to transit from one spawning site to another, these impacts should be minimised. We do not think of these in the marine environment very often because we attribute dispersal to larvae, but this focus on benthic adult or juvenile movement raises analogies to managing terrestrial animal connectivity.

Response to reviewers

Comments on revised ms:

General comments:

As usual, the revised manuscript is much improved. Much of the improvement is based on the added text that directly addresses the concerns of both reviewers. I find the revision more compelling and a clearer characterization of the system, the results, and the author's interpretations.

Thank you again for the helpful input our manuscript – we agree that the revised version is much improved as a result of the edits prompted by both reviewers' comments.

The revised manuscript, especially the added lines 265-268 and lines 318-321(!), makes clear that connectivity is only one important contributor to predator resilience. While I very much appreciate the emphasis on the contribution of connectivity to system resilience, I suggest the authors re-think the title "Spatial connectivity increases ecosystem resilience to an ongoing regime shift" to be more inclusive "Predation, spatial connectivity, and the local environment interact to increase ecosystem resilience to an ongoing regime shift". I say this because (1) it still reinforces the importance of spatial connectivity as a driver of ecosystem resilience, but (2) goes a step further to indicate that the connectivity effect interacts with other drivers (predation, temperature). It's an even greater advance than identifying the role of connectivity alone. Those looking for evidence of the importance of species interactions (predator-prey), spatial connectivity, and environmental conditions (temperate) and their interactions(!) will be drawn to the value of this article.

Very good point! To try to keep the title a bit shorter, we went with "Predation and spatial connectivity interact to shape ecosystem resilience to an ongoing regime shift".

I have the remaining recommendations:

I do think it is necessary to include the time series that the authors provided the reviewers with as supplementary information in the revised manuscript! What I find especially interesting, depending on the exposure of these sites to stickleback recruitment, is the 5 of 25 sites that persisted as predator dominated, suggesting resistance to invasion. If these 5 sites were subjected to stickleback recruitment, it would be strong evidence for resistance while the other sites exhibit greater vulnerability to transitioning to the stickleback dominated state. And the question is whether these 5 sites experienced greater predator connectivity than the others.

We agree that these time series are interesting, and helpful for better understanding the dynamics of the system. The five sites that maintain stable predator dominance would be good to monitor more closely in the future to see whether any stickleback actually shows up there in spring. Unfortunately we do not have the detailed local data to look at that at the moment, and we do not think that our predictions of "stickleback exposure" are precise enough when dealing with such a small dataset.

We have added the time series as Supplementary Fig. 7, and have added some discussion on the time series to the main text (L306-318).

Line 144: Delete comma after "predation/fishing"

The comma has been deleted (L114).

Line 205: “effect of on the absolute densities of predator juveniles...” Effect of connectivity?
Well spotted. This was meant in general, for any driver, and so we removed “of” (L167).

Line 270 (first paragraph of Discussion) While I appreciate the emphasis on connectivity to highlight its importance to resilience, this first sentence summarizing the drivers of resilience doesn’t mention what the authors identified as a crucial driver, predation. I would restate this to say that they found that interactions of predation, connectivity and the environment (temperature) all contributed critically to estimates of resistance/resilience of the predator-dominated state of the ecosystem. Then the rest of this paragraph describes those interactions.

Good point. We have adjusted this sentence (L215-218).

Line 280: In this section, simply make clear here that the vehicle of connectivity is the adult stage. “The movement of adult predatory fish between spawning areas allows for redistribution and recolonisation.” and “high predation pressure from top predators, most likely because there were too few adult predators to redistribute.” The revised Methods that includes information on larval dispersal is invaluable for clarifying why adult movement is focused on here. But it is good to clarify it in this section, rather than the reader having to read the Methods to understand that adult movement is the basis of connectivity here and why.

Good suggestions, we have added “adult” on L227 and L230.

Line 375: This new text is extremely valuable for teasing apart the relative influences of resistance and recovery (how I define resilience) as they contribute to resilience or persistence of the predator dominated state. As you know, there is an extensive literature on the importance of resistance on ecosystem invasibility and the mechanisms of resistance (e.g., species diversity and identity) versus capacity for recovery (e.g., changes in species recruitment rates) can be fundamentally different. This paragraph helps to address this.

We are happy to hear that the new text was helpful.

Line 384: Among the many potential management implications identified, I didn’t see one that truly reflects the role of connectivity, which is to ensure that connectivity corridors are maintained. If human activities can impact the quality of habitat required to transit from one spawning site to another, these impacts should be minimised. We do not think of these in the marine environment very often because we attribute dispersal to larvae, but this focus on benthic adult or juvenile movement raises analogies to managing terrestrial animal connectivity.

Very good point, we have added a line that highlights the importance of protecting important dispersal corridors (L336-338).